# A Temperature-Dependent Viscoplasticity Model for the Hot Work Steel X38CrMoV5-3, Including Thermal and Cyclic Softening under Thermomechanical Fatigue Loading

**DOI:** 10.3390/ma16030994

**Published:** 2023-01-21

**Authors:** Markus Schlayer, Marc Warwas, Thomas Seifert

**Affiliations:** 1Institute for Digital Engineering and Production (IDEeP), Offenburg University of Applied Sciences, Badstraße 24, 77652 Offenburg, Germany; 2Fraunhofer Institute for Mechanics of Materials IWM, Wöhlerstraße 11, 79108 Freiburg, Germany

**Keywords:** fatigue, strengthening mechanism, thermomechanical processes, cyclic loading, elastic–viscoplastic material, thermal stress

## Abstract

In this paper, a temperature-dependent viscoplasticity model is presented that describes thermal and cyclic softening of the hot work steel X38CrMoV5-3 under thermomechanical fatigue loading. The model describes the softening state of the material by evolution equations, the material properties of which can be determined on the basis of a defined experimental program. A kinetic model is employed to capture the effect of coarsening carbides and a new isotropic cyclic softening model is developed that takes history effects during thermomechanical loadings into account. The temperature-dependent material properties of the viscoplasticity model are determined on the basis of experimental data measured in isothermal and thermomechanical fatigue tests for the material X38CrMoV5-3 in the temperature range between 20 and 650 ${}^{\circ }$C. The comparison of the model and an existing model for isotropic softening shows an improved description of the softening behavior under thermomechanical fatigue loading. A good overall description of the experimental data is possible with the presented viscoplasticity model, so that it is suited for the assessment of operating loads of hot forging tools.

## 1. Introduction

Hot forging is an economically widespread process for achieving high degrees of deformations. Therefore, semifinished parts are heated above 1000 ${}^{\circ }$C and, for example, formed in a forging die [1]. In service, the hot forging tools are exposed to high cyclic mechanical and thermal loading conditions in addition to wear [2]. Although the temperature inside the tools is almost constant over the forming cycle after run-in of the tools [3], contact with the billet during forming and cooling of the hot work tools after each forging step results in high mechanically induced stresses as well as in thermally induced stresses due to the sharp temperature gradients in near-surface regions. Hence, thermomechanical fatigue (TMF) is a relevant damage mechanism for hot forging tools [4]. Because of their suitable mechanical properties like high levels of strength and hardness combined with sufficient ductility, martensitic hot work steels are often chosen as material for the tools nowadays.

To prevent early and unexpected failure of the tools due to the high cyclic thermal and mechanical loads, the tools are often designed by using finite-element calculations. It is state of the art to use a linear elastic analysis by comparing the equivalent von Mises stress with the yield strength of the material and the maximum principle stress with the tensile strength of the material [5]. However, the occurring stress in service can locally (e.g., in fillet radii and other notches) exceed the materials’ yield strength. Hence, appropriate plasticity models are required that can describe the cyclic mechanical behavior of the used martensitic hot work steels in the relevant temperature range from room temperature to around 600 ${}^{\circ }$C [6,7].

Generally, plasticity models for finite-element calculations of TMF loaded components are readily available, e.g., as presented in [8,9,10,11,12,13]. These models are usually based on the works of Chaboche [14,15,16] and contain isotropic and kinematic hardening as well as static recovery. A plasticity model with isotropic and kinematic hardening is e.g., used for the TMF assessment of a hot forging tool in [17]. Models for TMF assessment usually use viscoplastic formulations to account for time-dependent behavior at higher temperatures, as it is observed for martensitic hot work steels at temperatures above approximately 400 ${}^{\circ }$C [18]. However, these models do not account for softening phenomena that are, in particular, relevant for martensitic hot work steels.

### 1.1. Softening Mechanisms in Martensitic Steels

Softening is especially observed in the surface region of hot forging tools that are in contact with the semifinished parts. A reduction of hardness from initially 47 to 33 HRC was observed in [19] in a superficial region with a depth of 0.3 mm of a hot forging tool made from martensitic steel X40CrMoV5-1, being a preferred site for the nucleation and propagation of fatigue cracks. Continuous softening of the hot work steel 4Cr5MoSiV1 was observed in [20] in strain-controlled in-phase (IP) and out-of-phase (OP) TMF tests in the temperature range from 400 to 700 ${}^{\circ }$C. Due to softening, the stress levels decrease in the entire temperature range with an increasing number of cycles. Softening in the martensitic steels is attributed to two mechanisms, namely thermal softening controlled by temperature, e.g., analyzed in [21,22,23], and cyclic softening controlled by cyclic plasticity, e.g., analyzed in [24] and in the review article by Jilg [25].

Thermal softening is observed for temperatures above 400 ${}^{\circ }$C as the strengthening effect of secondary carbides that precipitate during the tempering process and lead to the Orowan mechanism decreases as a result of the coarsening of the carbides. The microstructure and its development at elevated temperatures is widely studied for the hot work steel X38CrMoV5-3 and comparable steels. Investigations with scanning electron microscopy (SEM) and transmission electron microscopy (TEM) are carried out for different tempering times in [26,27] showing temperature-dependent coarsening of carbides, as e.g., M${}_{23}$C${}_{6}$. In [22], the effect of thermal softening is correlated with carbide coarsening by aging experiments on basis of TEM micrographs. However, softening occurs faster under cyclic plastic straining at higher temperatures than would be expected from particle coarsening alone [21]. Hence, in addition to thermal softening, cyclic softening is relevant for martensitic hot work steels under fatigue loading. Cyclic softening is thereby attributed to a decrease of the dislocation density during cyclic plastic loading, e.g., shown in [28]. Findings from a high-resolution experimental analysis in [29] show that coarsening of carbides is not affected by the superimposed mechanical strain but solely caused by the thermal loading. Hence, thermal softening and cyclic softening can be treated as independent mechanisms.

### 1.2. Plasticity Models for Martensitic Steels Including Softening

For thermal softening, a relation to the microstructure is provided by using kinetic models describing the coarsening of the carbides of martensitic hot work steels, as e.g., in [27,30,31] where the mechanism of Ostwald ripening is considered and the LSW theory (originally published by Lifshitz [32] and Oriani [33]) is used and in [28,34] where the Johnson–Mehl–Avrami equation describes the tempering kinetics. Refs. [30,31] use nonisothermal tempering tests to validate the kinetic model by correlation with hardness measurements. Kinetic models are combined with Chaboche-type plasticity models e.g., in [35,36], so that the plasticity models are able to describe the effect of thermal softening on the mechanical behavior of the considered martensitic hot work steels. However, only isothermal loadings are used in these works so that no validation of the combined models for TMF loading, as relevant for this work, is given yet.

Based on low-cycle fatigue tests for the martensitic hot work steel 55NiCrMoV8 in the temperature range between 200 and 550 ${}^{\circ }$C, cyclic softening is modeled by using an analytical isotropic softening function for each temperature in [37]. These functions, however, cannot be used for TMF loading conditions because the history dependency of softening during nonisothermal cyclic loading is not reproduced. The same isotropic hardening law is applied in [38] in differential form (still lacking history dependency as relevant for TMF loadings) to formulate a nonisothermal model for application to TMF. Although TMF conditions are investigated experimentally for 55NiCrMoV7, no comparison of the evolution of softening during TMF between model and experiment is given. Moreover, both models do not consider the softening effect caused by coarsening of carbides. A similar approach with analytical isotropic-softening functions is used in [39], where softening under isothermal fatigue conditions of the ferritic–martensitic steel P91 is considered for temperatures up to 600 ${}^{\circ }$C, and in [40], where softening under isothermal fatigue of different steels including the martensitic steel G115 at 650 ${}^{\circ }$C is modeled. Both steels are used e.g., in power-generation applications and are susceptible to carbide coarsening during thermal loading which is not considered in these works. A multi-mechanism model using two inelastic strains and corresponding hardening variables are used to describe plasticity under isothermal fatigue including history-independent cyclic softening of 55NiCrMoV7 for a temperature range between 20 and 500 ${}^{\circ }$C in [41]. Their model is extended with temperature rate terms as relevant for nonisothermal loading in [42]. However, only model results are shown and experimental results for nonisothermal conditions for validation are not considered.

In [35], a kinetic model is introduced into the nonisothermal plasticity model with the history-independent isotropic cyclic-softening model presented in [38], so that both thermal and cyclic softening are considered in the plasticity model. In the combined model, it is assumed that the kinetic model describes coarsening as a purely thermal effect acting independent of cyclic mechanical straining. This assumption is in accordance with the findings of [29]. In the combined model, the material properties are dependent on an aging variable. The material properties are determined from isothermal fatigue tests. However, the developed plasticity model with thermal and cyclic softening is not validated with TMF loadings as they are in the focus for this work.

### 1.3. Aims and Structure of the Paper

The preceding sections show that the basic mechanisms of thermal and cyclic softening in martensitic steels are quite well understood. However, the developed plasticity models do not account for the history dependency of softening and are not validated for TMF conditions based on experimental data. Hence, it is the aim of this work to develop a temperature-dependent viscoplasticity model including history-dependent thermal and cyclic softening under TMF loading. By using the known softening mechanisms as a basis, a phenomenological model is developed, the material properties of which can be determined on the basis of isothermal tempering hardness curves and a defined set of isothermal and thermomechanical fatigue tests.

The martensitic hot work steel X38CrMoV5-3 widely used as a material for hot forming tools is considered in this work. Because a database for the assessment of softening under thermomechanical loading conditions is not yet available for this steel, an experimental program is designed for investigation of softening effects on temperature-dependent cyclic viscoplasticity comprising isothermal and thermomechanical fatigue tests in the temperature range from room temperature to 650 ${}^{\circ }$C and different heat treatments. The database serves for the development of a combination of a kinetic model and a viscoplasticity model for which a new isotropic cyclic-softening variable and the corresponding evolution equation for TMF is derived. The kinetic model as well as the corresponding material properties for X38CrMoV5-3 (determined from isothermal tempering hardness curves) are taken from [30]. For the determination of the material properties of the viscoplasticity model with the new cyclic softening variable, a stepwise strategy is proposed that can serve as guideline to apply the model to other martensitic steels. The model is validated by using TMF loads relevant for the application of the model in the design of hot forging tools to avoid unplanned failures. Against the background of the known mechanisms, no microstructural investigations are used for validation purposes in this work.

The paper is structured as follows: In the next section, the material and experiments are described. In Section 3, the viscoplasticity model including thermal softening and a new cyclic softening model is introduced. Furthermore, the strategy to determine the material properties is presented. The experimental results and the model results using the determined material properties are presented in Section 4. The results are discussed in Section 5, and the work is concluded in Section 6.

## 2. Materials and Methods

In this section, the test procedure and the performed tests are presented. Section 2.1 contains the information on the investigated material and the preparation of the specimen for the fatigue tests. The performed fatigue tests as well as the test equipment are presented in Section 2.2.

### 2.1. Material

In this work, the martensitic tool steel X38CrMoV5-3 is investigated. Table 1 shows the chemical composition in accordance with DIN EN ISO 4957 [43] and the chemical composition of the specimen determined by optical emission spectrometry and a shutter diameter of 12 mm. The molybdenum content falls below the specifications of DIN EN ISO 4957. However, the value is within the permissible deviations due to product analysis [43].

The specimen for the fatigue tests are tempered in three steps to reach a martensitic material matrix with fine grains [44,45]. The hardness after tempering is 54 HRC. The specimen are manufactured as round bars with a length of 125 mm and a gauge diameter of 5 mm. After tempering and turning the specimen to their final size, some of the specimen is heat treated (HT) for evaluating different stages of aging in the fatigue tests. Table 2 shows the different tempering procedures with the assigned notations HT0, HT1, and HT2.

### 2.2. Isothermal and Thermomechanical Low-Cycle Fatigue Tests

For the determination and validation of the material properties of the viscoplasticity model, isothermal and thermomechanical fatigue tests are carried out with specimen of HT0, HT1, and HT2. These heat treatments are chosen in order to establish a consistent data basis with previous works [30] and because there is a clear influence of the heat treatment on the mechanical material behavior under these conditions.

All tests are performed on an electromechanical fatigue testing system of Instron (Model 1362) with a moving path of ±50 mm. The tests are performed in strain control by measuring the strain with a ceramic extensometer manufactured by Epsilon. The range of the extensometer is ±500 μm. The temperature is prescribed and controlled by inductive heating and measured with two thermocouples wrapped around the specimen in the gauge length. The induction system is manufactured by Ambrell with a capacity of 10 kW. The thermocouples are from SensyMIC and of type K with a temperature range from −40 to 1200 ${}^{\circ }$C. The force is measured by a load cell and converted into stress. The load cell is from Instron (model 2518-100) with a static load capacity of 100 kN and a dynamic load capacity of 50 kN in tension and compression.

The isothermal fatigue tests use so-called complex low-cycle fatigue tests (CLCF) in which during a complex strain history in the first cycles different strain rates ($10^{-3}$, $10^{-4}$, $10^{-5}$ 1/s) are used and dwell times of 1800 s are applied under tensile and compressive load (see Figure 1). This complex part of the test allows a direct evaluation of the time-dependent material behavior like strain rate dependency and stress relaxation in one test [46]. Subsequently, in the cyclic part of the test following the complex part, the specimen is tested with a constant strain rate of $10^{-3}$ 1/s with a constant mechanical strain amplitude until failure of the specimen defined by a 5% drop of the maximum stress in tension with respect to ISO 12106. The CLCF tests are carried out at 20, 400, 500, 600, and 650 ${}^{\circ }$C for HT0, HT1, and HT2. The strain amplitude in the complex part of the tests is 0.008 and 0.01, and it is 0.01 in the cyclic part for all temperatures and heat treatments of the tests.The strain ratio for all isothermal tests is −1 (fully reversed). Table 3 gives the corresponding cycles to failure.

The results for 20 and 650 ${}^{\circ }$C are shown (already together with the modelling results) in Figure 2 and Figure 3 (complex parts) and Figure 4 and Figure 5 (complete experiments). Figure 2 shows for 20 ${}^{\circ }$C higher stress values for HT0 than for HT1 and HT2 due to thermal softening of the material during heat treatment. For all tests at 20 ${}^{\circ }$C, only a minor influence from the viscous material behavior is observed. Cyclic softening hardly occurs. For 650 ${}^{\circ }$C (Figure 3), the stresses within the cycles are lower than in the tests at 20 ${}^{\circ }$C. Furthermore, the difference in stress between HT0, HT1, and HT2 is small and disappears in the last cycles. Softening is significant for 650 ${}^{\circ }$C throughout the CLCF tests.

Two types of TMF tests are carried out as shown in Figure 6 with the mechanical strain and temperature history of the applied loading cycles with a cycle time of 180 s. A strain amplitude of 0.005 is applied for the IP TMF tests for all cycles to failure and a strain amplitude of 0.01 for the OP TMF tests. In OP TMF tests, temperature and mechanical strain have a phase shift of 180${}^{\circ }$. In the IP TMF tests, the temperature is applied without phase shift with mechanical strain. The temperature within a TMF cycle varies between 200 and 650 ${}^{\circ }$C. The temperature rate is 5 K/s and constant over the loading cycles and experiments. An overview of the number of cycles to failure of the TMF tests is given in Table 4.

The measured stresses in the TMF tests are shown (already together with the modelling results) in Figure 7 for the IP TMF tests and in Figure 8 for the OP TMF tests. Unfortunately, the stress-strain data for the OP TMF tests is not complete for HT1 due to difficulties with testing equipment, although the number of cycles to failure in Table 4 is correct. Figure 7 shows higher (negative) stress values in compression than in the tensile part of the cycles for HT0, HT1, and HT2. At the end of the IP TMF tests, the stress levels of HT0, HT1, and HT2 are on the same level due to softening of HT0 and HT1. For IP TMF tests in Figure 7, softening occurs for HT0 and HT1 whereas the HT2 tests show only minor influence on softening. However, the number of cycles to failure is significant smaller for the IP TMF tests for HT2 than for HT0 and HT1. Figure 8 shows the stress values of the OP TMF tests. The highest stress values for HT0, HT1, and HT2 are measured at the lowest temperature at which the highest strain is applied. Thermal and cyclic softening occurs for all heat treatments in the OP TMF tests. Thermal softening is shown by different stress values of HT0, HT1, and HT2 in the initial cycles of the OP TMF tests. Cyclic softening is shown by decreasing stress values for HT0, HT1, and HT2 with increasing number of cycles. The softening range of HT0 is the highest compared to HT1 and HT2. At the end of the tests, the measured stress of HT0 and HT2 decrease to the same stress values during compression.

## 3. The Viscoplasticity Model Including Thermal and Cyclic Softening

In this section, the temperature-dependent viscoplasticity model is presented. The base model is presented in Section 3.1 together with the main equations. Section 3.2 deals with the new cyclic softening model that considers the history of softening during nonisothermal cyclic loading. The kinetic model describing thermal softening is presented in Section 3.3.

### 3.1. The Base Model

A cyclic viscoplasticity model based on the works of Chaboche [14,15] is used as base model to include the effects from softening in Section 3.2 and Section 3.3. The cyclic viscoplasticity model is presented in its uniaxial formulation because it is developed based on uniaxial material tests.

The stress σ is calculated from Young’s modulus and elastic strain $\varepsilon^{\mathrm{el}}$ as
(1)$$\sigma =E\cdot \varepsilon^{\mathrm{el}}=E\cdot \left(\varepsilon -\varepsilon^{\mathrm{th}}-\varepsilon^{\mathrm{vp}}\right),$$
whereby the elastic strain can be computed from the total strain ε, thermal strain $\varepsilon^{\mathrm{th}}$ and the viscoplastic strain $\varepsilon^{\mathrm{vp}}$. The viscoplastic strain is computed by integration of the flow rule
(2)$${\dot{\varepsilon }}^{\mathrm{vp}}={\dot{\bar{\varepsilon }}}^{\mathrm{vp}}\cdot \frac{\sigma -\alpha }{\left|\sigma -\alpha \right|}.$$
${\dot{\bar{\varepsilon }}}^{\mathrm{vp}}$ is the equivalent visoplastic strain rate. The flow direction is computed by the stress σ and the backstress α, the latter describing kinematic hardening of the material.

The equivalent viscoplastic strain rate is determined by the power-law
(3)$${\dot{\bar{\varepsilon }}}^{\mathrm{vp}}={\dot{\bar{\varepsilon }}}_{0}^{\mathrm{vp}}\cdot {\langle \frac{|\sigma -\alpha |-R_{p}}{K}\rangle }^{n}.$$ The numerator represents the viscous overstress and the denominator *K* is a temperature-dependent material property that defines the viscous overstress obtained for the reference strain rate ${\dot{\bar{\varepsilon }}}_{0}^{\mathrm{vp}}$. The exponent *n* is a material property. *K* and *n* quantify the time-dependent behavior of the material (as e.g., stress relaxation and rate dependency). In Section 3.3, these material properties are used to include the effect of thermal softening. 〈〉 are the Macaulay brackets so that viscoplastic yielding only takes place in case of positive overstress.

The viscoplasticity model considers kinematic hardening via the backstress α representing the internal residual stress fields producing the Bauschinger effect under cyclic plastic loadings. The evolution of the backstress is described by the evolution equation based on the Frederick–Armstrong hardening law [47]: (4)$$\dot{\alpha }=C\cdot {\dot{\varepsilon }}^{\mathrm{vp}}-\gamma \cdot {\dot{\bar{\varepsilon }}}^{\mathrm{vp}}\cdot \alpha -R\cdot \alpha +\frac{\partial C}{\partial T}\cdot \frac{\dot{T}}{C}\cdot \alpha .$$ The kinematic hardening modulus *C* and the dynamic recovery parameter γ are temperature-dependent material properties to describe the nonlinear (exponential) hardening curve with the first two terms in the evolution equation. The material property $C_{\infty }=C/\gamma$ defines the saturation value. The third term describes static recovery of kinematic hardening, i.e., recovery of hardening with time usually occurring at higher temperatures [14]. Static recovery is quantified by the material property *R*. In the temperature rate term (last term in the evolution equation, *T* is the temperature) derived from thermodynamics, only the temperature dependency of the hardening modulus *C* is accounted for [15]. Thus, only for a constant (temperature-independent) value of γ, the backstress is bounded for nonisothermal conditions by the current saturation value $C_{\infty }$. In Section 3.3, the material properties *C*, γ (or $C_{\infty }$) and *R* will also be used to include the effect of thermal softening in the model.

The isotropic strength of the material is described by the variable $R_{p}$ in Equation (Equation 3). If no isotropic static recovery is considered, isotropic hardening or softening often is accounted for by using the analytical isotropic hardening law (e.g., [48])
(5)$$R_{p}=R_{e}+H\cdot {\bar{\varepsilon }}^{\mathrm{vp}}+Q_{\infty }\cdot \left(1-e^{-b\cdot {\bar{\varepsilon }}^{\mathrm{vp}}}\right).$$
$R_{e}$ is the initial yield stress. Hardening or softening is controlled by the accumulated viscoplastic strain ${\bar{\varepsilon }}^{\mathrm{vp}}$. *H* is the hardening modulus of the linear part (second term) and $Q_{\infty }$ is the saturation value of the exponential isotropic hardening function (third term) and *b* is the corresponding dynamic recovery parameter. All these material properties might depend on temperature. This analytical hardening law is used in [37] to model two-stage softening of the hot work steel 55NiCrMoV8 consisting of a primary rapid load decrease, followed by secondary linear cyclic softening. Hence, *H* and $Q_{\infty }$ are identified as negative values. Under TMF loading, however, this analytical model is not able to reproduce history effects as observed in the TMF tests in this work (Section 2.2): softening of the material that occurred at higher temperatures also affects the behavior when subsequently cooled to a lower temperature. The use of the differential form of this model, as done in [35,38] to model cyclic softening of 55NiCrMoV7, cannot eliminate this deficiency of the model. Hence, an appropriate isotropic cyclic-softening model that describes history effects under TMF loading appropriately is derived in the next subsection.

### 3.2. The New Cyclic Softening Model

For the description of history effects in cyclic softening under TMF loading conditions, a softening variable *s* is introduced in the model that is defined in the range from 0 to 1. With s=0, the initial unsoftened material is determined, whereas s=1 determines the fully softened state. The evolution of the softening variable is described with an evolution equation that is inspired by Cailletaud [49] who derived an aging variable and its evolution for TMF loaded age-hardenable aluminum alloys. Their aging variable, however, evolves as a function of time and temperature (via the material properties). As cyclic softening of hot work steels under TMF loading was found to be driven by the accumulated plastic strain [38], the evolution equation of the ageing variable of Cailletaud [49] is adapted in this work to describe the plasticity-controlled evolution of the softening variable by
(6)$$\dot{s}=\langle \frac{s_{\infty }-s}{p_{0}}\rangle \cdot {\dot{\bar{\varepsilon }}}^{\mathrm{vp}}.$$ The material property $0\leq s_{\infty }\leq 1$ defines the maximum value that the softening variable can attain at a certain temperature, i.e., $s_{\infty }$ is temperature dependent. Due to the Macauley brackets, an evolution of *s* is only possible if $s<s_{\infty }$ is currently true. Hence, at lower temperatures, where hardly any cyclic softening is observed and $s_{\infty }$ is small, there will be no further softening occurring. However, the softening state attained during prior loading at higher temperatures is still maintained. The material property $p_{0}$ controls the softening rate and is temperature dependent as well.

With the new softening variable, isotropic softening of the material can be introduced in the viscoplasticity model. Similar to the model in [49], the isotropic strength of the material described by $R_{p}$ now depends on the softening variable
(7)$$R_{p}=R_{0}+R_{s}\cdot \left(1-s\right).$$
$R_{0}$ quantifies the temperature dependent intrinsic yield strength as the minimum value of $R_{p}$ for a fully softened material state with s=1. $R_{s}$ gives the temperature-dependent amount of strength that can get lost by softening. Hence, initially for the unsoftened material with s=0, the isotropic strength is $R_{p}=R_{0}+R_{s}$. Because the material property *K* defines the viscous overstress that also contributes to the strength of the material and can change during plastic loading (as e.g., function of the accumulated viscoplastic strain [14]), the same approach as in Equation (Equation 7) is used to describe the dependency of *K* on the softening variable
(8)$$K=K_{0}+K_{s}\cdot \left(1-s\right).$$
$K_{0}$ and $K_{s}$ are again temperature-dependent material properties quantifying the intrinsic and the softening contribution. Both $R_{s}$ and $K_{s}$ can only be determined from nonisothermal tests (Section 4) because they quantify the current loss of strength at a certain temperature apparent from a previous high-temperature loading.

For isothermal conditions, the evolution Equation (Equation 6) can be solved, analytically yielding
(9)$$s=s_{\infty }\cdot \left(1-e^{-\frac{{\bar{\varepsilon }}^{\mathrm{vp}}}{p_{0}}}\right),$$
and, thus, an exponential increase of *s* with saturation value $s_{\infty }$ and transition constant $p_{0}$. In this case, insertion of Equation (Equation 9) into Equation (Equation 7) gives
(10)$$R_{p}=R_{0}+R_{s}-s_{\infty }\cdot R_{s}\left(1-e^{-\frac{{\bar{\varepsilon }}^{\mathrm{vp}}}{p_{0}}}\right),$$
which corresponds to the analytical isotropic hardening law of Equation (Equation 5) neglecting the linear hardening term. One finds $R_{0}+R_{s}=R_{e}$ as the initial yield stress and the product $s_{\infty }\cdot R_{s}=-Q_{\infty }$ describing the maximum possible loss in strength as well as $p_{0}=1/b$. Analogously,
(11)$$K=K_{0}+K_{s}-s_{\infty }\cdot K_{s}\cdot \left(1-e^{-\frac{{\bar{\varepsilon }}^{\mathrm{vp}}}{p_{0}}}\right)$$
for isothermal conditions. Hence, $s_{\infty }$ is determined from isothermal fatigue tests as the portion of $R_{s}$ and $K_{s}$ that is lost in isothermal fatigue tests and $p_{0}$ quantifies the softening rate. In Section 3.3, $R_{0}$, $R_{s}$, $K_{0}$, and $K_{s}$ are additionally defined to be affected by thermal softening.

A comparison of the model results with the analytical isotropic hardening law of Equation (Equation 5) and the new softening model proposed in this section is presented in Figure 7 and Figure 9 for the IP TMF tests and in Figure 8 and Figure 10 for the OP TMF tests.

### 3.3. The Kinetic Model Describing Thermal Softening

Thermal softening of martensitic hot work steels is a result of coarsening of strengthening carbides, which can be described by kinetic models. In this work, the kinetic model developed in [36] is used. There, the material properties for the hot work steel considered in this work are already determined based on isothermal and nonisothermal tempering hardness curves for the considered martensitic tool steel X38CrMoV5-3. The normalized carbide size *z* with value 1 corresponding to the HT0 material is introduced in this work as qualitative measure describing the temperature-dependent coarsening of the strengthening carbides with time and, hence, characterizing the current strength.

The kinetic model assumes that one carbide species exists that dominantly determines the mechanical properties. It describes the coarsening of this carbide species by Ostwald ripening by which larger carbides grow thermally activated at the expense of smaller ones keeping the volume fraction of the carbides constant. Hence, the coarsening rate of the carbides is given by
(12)$$\dot{z}=\frac{k}{z^{2}}\mathrm{with}k=\frac{k_{1}}{T}\cdot e^{-\frac{Q}{R_{g}\cdot T}}.$$ Here, *T* is the absolute temperature and $R_{g}$ is the universal gas constant. For the martensitic hot work steel X38CrMoV5-3 and the normalized normalized carbide size considered in this work, the activation energy *Q* is 340.2 kJ/mol and the coarsening constant $k_{1}$ is 3.336 $\times \;10^{20}$ K/min [36]. Integration of Equation (Equation 12) for isothermal conditions (as present during heat treatment) gives $z^{3}-z_{0}^{3}=3kt$ with time *t*. With the given material properties, the normalized carbide size for the HT1 and the HT2 material (Table 2) is 1.406 and 3.955, respectively.

The results of the CLCF tests for the HT0, HT1, and HT2 material show that the material behavior significantly depends on the heat treatments, i.e., on the coarsening state of the carbides. Thus, the material properties of the viscoplasticity model, namely *K* and *n* from Equation (Equation 3), and *C*, γ and *R* from Equation (Equation 4), as well as the material properties of the softening model, namely $p_{0}$ and $s_{\infty }$ from Equation (Equation 6) and $R_{0}$, $R_{s}$, $K_{0}$ and $K_{s}$ from Equations (Equation 7) and (Equation 8) are assumed to also be dependent on the carbide size *z*. The functional dependency on carbide size is addressed in Section 4.2.

To account for the additional dependency of the kinematic hardening modulus *C* on *z*, the following contribution,
(13)$$\frac{\partial C}{\partial z}\cdot \frac{\dot{z}}{C}\cdot \alpha ,$$
is added to the evolution equation for the backstress given in Equation (Equation 4).

## 4. Results

In this section, the material properties of the viscoplasticity model with thermal and cyclic softening (Section 3) are determined based on the experimental data (Section 2). The material properties depend on temperature and the carbide size, i.e., the heat treatment. In the model, mechanical loading does not affect thermal softening, so that the kinetic model can be applied with its already given material properties and the functional dependencies of the material properties of the viscoplasticity model on the carbide size *z* can be derived from the CLCF data available for HT0, HT1, and HT2 at 20, 400, 500, 600, and 650 ${}^{\circ }$C. Moreover, the material properties of the cyclic softening model can be determined independently from the model for thermal softening. However, the determination of some material properties of the cyclic-softening model requires the data from TMF tests. Hence, the following strategy is defined for the determination of the material properties.

**Step 1**:Determination of individual material properties on the basis of the complex part of the CLCF tests neglecting thermal and cyclic softening.The initial loading cycles of the CLCF tests are considered, assuming that cyclic softening is not significant during the initial cycles. From that data, the temperature-dependent material properties are determined for HT0, HT1, and HT2 individually.**Step 2**:Determination of functional dependencies of the material properties from thermal and cyclic softening on the basis of the full CLCF and TMF tests.The material properties for the cyclic-softening model (Section 3.2) are determined from all cycles of the CLCF and TMF tests and the functional dependencies of the material properties on normalized carbide size *z* are derived. All cycles of the CLCF and TMF tests are computed with the viscoplasticity model including thermal and cyclic softening.

In the following, the determination of the material properties and their functional dependencies within the two steps is addressed. Subsequently, a comparison of calculations with the new cyclic-softening model and the analytical isotropic-softening model is presented in Section 4.3. For better comparison, the new cyclic-softening model from Section 3.2 is designated as new softening model and the isotropic-softening model from Equation (Equation 5) as standard softening model.

### 4.1. Step 1: Material Properties for the Initial Behavior Neglecting Thermal and Cyclic Softening

In this step, the material properties of the viscoplasticity model (base model) are determined on the basis of the first 11 cycles of the CLCF tests at 20, 400, 500, 600, and 650 ${}^{\circ }$C for the HT0, HT1, and HT2 material individually. The material properties are determined without dependency on the carbide coarsening *z*, i.e., without dependency on thermal softening. Furthermore, cyclic softening is neglected in the model, i.e., s=0 and $\dot{s}=0$, so that $R_{p}$ and *K* are treated as individual material properties. For the determination, an “experience-based approach” is used to obtain physically reasonable temperature dependencies of the material properties (e.g., an increase of the viscosity and a decrease of strength with increasing temperature), i.e., the material property values are determined manually from experimental evidence (no optimization algorithm is used). Subsequently, the material properties determined from the isothermal tests are validated for nonisothermal conditions by comparison of results obtained with the viscoplasticity model for the TMF tests with the corresponding experimental results.

The material properties determined in this step individually for HT0 (z=1), HT1 (z=1.406), and HT2 (z=3.955) are shown in Figure 11 together with the functional dependency on the carbide size derived in step 2. The values of the individual material properties of step 1 that do not depend on softening are listed in the Appendix A. Young’s modulus is determined from the visible linear elastic loadings and unloadings during a loading cycle. Young’s modulus is temperature dependent; however, no dependency on carbide size and softening is observed. The material properties related to rate dependent behavior *K* and *n* for ${\dot{\bar{\varepsilon }}}_{0}^{\mathrm{vp}}=$1/s, and the material properties related to plasticity and hardening $R_{p}$, *C*, $C_{\infty }$ and *R* are determined, such that a reasonable description of the initial cycles in the CLCF tests at each temperature is achieved.

As an example, the results for 20 and 650 ${}^{\circ }$C are shown in Figure 2 and Figure 3. Each figure shows the experimentally measured stress (symbols) and the calculated stress (lines) as function of time or mechanical strain for the CLCF tests and the material states HT0, HT1, and HT2. In Figure 2, the model results at 20 ${}^{\circ }$C are in good accordance with the test data for HT0, HT1, and HT2. Time-dependent material behavior is hardly observed.

Figure 3 presents the results at 650 ${}^{\circ }$C. In the experiments, softening already occurs during theses first loading cycles. Hence, when neglecting softening in the viscoplasticity model, the material properties related to hardening can only be determined on the basis of the first cycle and the material properties related to viscous behavior can be estimated so that significant rate dependency and stress relaxation occurs appropriately.

The deviation between the model and test results in the CLCF tests at 650 ${}^{\circ }$ is reduced in the following step (Section 4.2) by combining the material properties with carbide size and enable cyclic softening. Figures showing further test temperatures (400, 500, and 600 ${}^{\circ }$C) are available as Appendix A. The results of the IP TMF tests are shown together with the model results in Figure 12 (left) and for the OP TMF tests in Figure 12 (right). Each figure shows the experimentally measured stress (symbols) and the calculated stress (line) as stress-temperature hysteresis loops of the first three cycles for the material states HT0, HT1 and HT2. At 200 ${}^{\circ }$C, the model underestimates the compressive stresses in the IP TMF tests while it overestimates the tensile stresses in the OP TMF tests. Overall, a good description of the three IP and OP TMF cycles is achieved with the viscoplasticity model and the material properties determined in step 1.

### 4.2. Step 2: Material Properties for the Full Behavior Including Thermal and Cyclic Softening

In this step, the functional dependencies of material properties on normalized carbide size (thermal softening) and the material properties related to cyclic softening are determined. Therefore, the individual material properties of step 1 (previous section) are maintained unchanged in this step.

Cyclic softening is accounted for in the viscoplasticity model via $R_{p}$ and *K* being dependent on the softening variable *s* as shown in Equations (Equation 7) and (Equation 8). The material properties $s_{\infty }$ and $p_{0}$ are determined from softening during isothermal fatigue tests. $R_{s}$ and $K_{s}$ are determined such that a good agreement between model and experimental data is obtained regarding softening during TMF tests. The assumption of constant ratios $R_{p}/K=R_{s}/K_{s}=R_{0}/K_{0}$ has proven to be helpful and appropriate in this step.

The dependency of the material properties on heat treatment and, thus, thermal softening is shown in Figure 11. Hence, a dependency on the normalized carbide size *z* is assumed for the following material properties: $R_{0}$ and $R_{s}$ (describing the carbide size-dependent $R_{p}$), $K_{0}$ and $K_{s}$ (describing the carbide size-dependent *K*) as well as *n*, *C*, $C_{\infty }$, $s_{\infty }$ and $p_{0}$. Different functional dependencies are defined. The functional dependency
(14)$$P\left(T,z\right)=P_{i}\left(T\right)+\frac{k_{P}\left(T\right)}{z}$$
is used for a material property *P* that decreases with coarsening of the carbides. $P_{i}$ is the intrinsic value, and $k_{P}$ is a constant, both being temperature dependent. This approach is especially reasonable for material properties related to the strength because the 1/r-dependency is a result of the Orowan mechanism. It applies to the following material properties in this work: $R_{0}$, $R_{s}$, *C*, $C_{\infty }$, *n*, and $s_{\infty }$. The functional dependency
(15)$$P\left(T,z\right)=P_{i}\left(T\right)+k_{P}\left(T\right)\cdot z$$
is used for a material property *P* that shows approximately linearly increasing values with carbide coarsening. $P_{i}$ is the intrinsic value and $k_{P}$ is a constant, both being temperature dependent. It applies to the following material properties in this work: $K_{0}$, $K_{s}$, and $p_{0}$. Finally, static recovery due to climb of dislocations is a diffusion based and, thus, thermally activated process. Significant recovery is observed for test temperatures higher than 500 ${}^{\circ }$C. Therefore, an Arrhenius equation is used to describe temperature dependency of the material property related to static recovery: (16)$$R\left(T,z\right)=A\cdot z\cdot e^{-\frac{E_{A}}{R_{g}\cdot T}}.$$ The factor *A* and the activation energy $E_{A}$ are assumed to neither depend on temperature nor on carbide size. The constants $P_{i}$, $k_{P}$, *A* and $E_{A}$ are determined such that the respective functional dependency results in a good description of the material properties that are determined for the different HTs in step 1 individually.

The functional dependencies resulting from Equations (Equation 14) to (Equation 16) in this step are given together with the individual determined material properties from step 1 in Figure 11. The determined functional dependencies are shown with the lines in Figure 11 and are in good agreement with the individual determined material properties. The material properties of step 2 compiled for the heat treatments HT0, HT1 and HT2 are available as Appendix A. Figure 4 shows the stress measured in the CLCF tests as symbols for three heat treatments at 20 ${}^{\circ }$C and Figure 5 at 650 ${}^{\circ }$C, respectively. The model results are illustrated with solid lines. In the cyclic part of the tests presented in Figure 4 (left) only the points of strain reversal are shown. Figures showing further test temperatures (400, 500, and 600 ${}^{\circ }$C) are available as Appendix A.

Cyclic softening is hardly occurring at 20 ${}^{\circ }$C. However, thermal softening is well described by the material properties including functional dependencies on carbide size; hence, the tests with HT0, HT1, and HT2 are described well by the model. The results of the CLCF tests for 650 ${}^{\circ }$C are presented in Figure 5. A good description of the tests is obtained with the viscoplasticity model. Thermal as well as cyclic softening are acting at 650 ${}^{\circ }$C. In the end of the HT2 tests, the stress drops rapidly because a macroscopic fatigue crack is now present.

The results of the TMF tests are shown together with the model results in Figure 7 and Figure 9 for the IP TMF tests and in Figure 8 and Figure 10 for the OP TMF tests. The stress-time diagrams are shown in Figure 7 and Figure 8, the stress–temperature hysteresis loops in Figure 9 and Figure 10—both on the left. Only the points of strain reversal are shown in the stress-time diagrams. Model results are shown for the standard softening model (standard sft, right figures) and the new softening model (new sft, left figues), respectively. The comparison of both models is addressed in Section 4.3. A good description of the IP TMF tests is achieved with the viscoplasticity model with the new softening model for HT0, HT1, and HT2. In particular, the softening range at low temperatures (e.g., 200 ${}^{\circ }$C) is well described by the model. However, the compressive stress is underestimated by the model in the first cycles for the HT1 material.

The stress-temperature hysteresis loops show good correlation of the model and experimental results. The largest cyclic softening is observed in the HT0 test. The OP TMF results for HT0, HT1, and HT2 are shown in Figure 8 and Figure 10. A fast drop in tensile stress is shown for the OP TMF test with HT0 material as well as in the model results. Therefore, the softening rate of the model results is in good correlation with the tests when using the transition constants for cyclic softening $p_{0}$ determined from isothermal tests. In the stress–temperature hysteresis loops in Figure 10 (left), the minimum stress values occur for the OP TMF tests at 600 ${}^{\circ }$C as also observed in the test. Overall, a good description of the OP TMF tests for the HT0, HT1, and HT2 material is achieved with the viscoplasticity model of step 2. The proportions of thermal and cyclic softening computed with the viscoplasticty model with the new cyclic-softening model are compared in Figure 13 for the HT0 material. Therefore, the tests are computed with different settings. First, thermal and cyclic softening are disabled (s=0 and $\dot{s}=0$; z=1 and $\dot{z}=0$). Furthermore, thermal and cyclic softening are enabled individually, and, finally, thermal as well as cyclic softening are enabled (full softening). This is possible because thermal and cyclic softening are determined independently in step 2. Figure 13 shows the tensile stresses at the strain-reversal points for 20 ${}^{\circ }$C (left) and 650 ${}^{\circ }$C (right), both with the HT0 material. The tensile stresses are shown versus the cycles. All stresses are subtracted from the results without softening and, thus, only softening of the different model definitions is shown. For 20 ${}^{\circ }$C, only cyclic softening occurs. No thermal softening takes place because of the coarsening constant that prevents thermal softening at 20 ${}^{\circ }$C. Therefore, full softening corresponds to cyclic softening. Overall, softening at 20 ${}^{\circ }$C is hardly occurring, which is quantified by the softening of in total 70 MPa. The gaps observed for the stress values are due to the hold times in the complex part of the tests. For 650 ${}^{\circ }$C, thermal as well as cyclic softening are acting. The superposition of thermal and cyclic softening leads to the full softening (black). It is shown that thermal softening is acting predominantly in the complex part (cycles 1 to 12) while cyclic softening is also occurring in the cyclic part of the tests. However, the saturation of thermal softening after the complex part can be explained with the time as the duration of the cycles in the complex part is longer than in the cyclic part (e.g., due to dwell times and lower strain rates in the complex part; see Figure 1). In comparison to 20 ${}^{\circ }$C the softening is significantly higher in absolute values and especially when referring it relatively to the strength at each temperature.

### 4.3. Comparison of TMF Results with Standard- and New Cyclic-Softening Model

In this section, a comparison of the results calculated with the viscoplasticity model of Section 3.1 by using the new history dependent cyclic softening model and the standard history independent model is shown. To this end, the material properties of step 2 are used, and the standard model is defined by evaluation of the analytical expressions for $R_{p}$ and *K* given in Equations (Equation 10) and (Equation 11) rather than the history-dependent forms for nonisothermal loading described by Equations (Equation 6) to (Equation 8). For isothermal loadings, both models give identical results. Hence, only results for TMF loading are considered.

The results for the IP TMF tests with the HT0, HT1 and HT2 material are shown in Figure 7 (stress-time diagrams) and Figure 9 (stress–temperature hysteresis). The stress values computed with the new softening model are shown on the left and the stresses computed with the standard model for isotropic softening are shown on the right side in both figures. Only the points of load reversal are shown in the stress–time diagrams. The largest difference between new model and standard model is shown for the HT0 material. Although the stresses calculated with the new softening model (left) show good accordance with the test data, the stresses calculated with standard softening model (right) overestimate the measured compressive stresses. In the IP TMF tests, the highest compressive stresses occur at the lowest temperature within a cycle. The stress-temperature hysteresis in Figure 9 shows that the softening at lower temperatures is underestimated by the standard softening in TMF tests for the HT0 material. This is also present for the HT1 material. However, for the HT2 material only minor differences in stress values are shown because the material is already softened. Similarly, the results of the OP TMF tests with the HT0, HT1, and HT2 material are shown in Figure 8 and Figure 10. Here, the highest stress values occur at the lowest temperatures and, thus, in tensile loading. The OP TMF stress-time diagrams in Figure 8 show good accordance with the softening behavior shown in the experimental results. The tensile stresses are overestimated by the standard model for the HT0 and HT1 material. As the HT2 material is already softened due to the heat treatment in advance of the experiments, only minor differences in the softening behavior between the new model and the standard model are observed. Furthermore, the impact of the softening model is compared on the basis of the calculated accumulated plastic strain. Therefore, Figure 14 shows the accumulated viscoplastic strain versus time for IP TMF (left) and OP TMF (right) tests with the HT0 material. In all IP TMF and OP TMF tests, the accumulated plastic strain increases faster in case of the new softening model, which describes the TMF data better than the standard model. Plastic deformations are underestimated with the standard model. This can cause underestimation of damage and, thus, overestimation of the lifetime of engineering components. Therefore, the new cyclic softening model has a significant impact on the evaluation of engineering components under TMF loading and their lifetime calculation. Hence, an improvement in the description of the experimental stress results due to the new softening model is obtained for TMF loading conditions.

## 5. Discussion

In this work, a viscoplasticity model for the hot work steel X38CrMoV5-3 (1.2367) is developed that considers effects from thermal and cyclic softening under thermomechanical loading conditions. CLCF and TMF tests are carried out to determine the time- and temperature-dependent plasticity and the softening behavior of the hot work steel to determine the material properties for the viscoplasticity model. A new cyclic softening model is introduced that captures history effects that are relevant to describe softening observed in TMF tests. Overall, a good description of the stress-strain behavior is obtained with the proposed viscoplasticity model.

### 5.1. Experimental Data and Material Properties

The results of the CLCF tests carried out at 20, 400, 500, 600, and 650 ${}^{\circ }$C with specimen with the heat treatments HT0, HT1, and HT2 show that strength decreases significantly with increasing tempering time and temperature. The reduction of strength due to thermal softening of the hot work steel is associated with an increase of ductility [50]. As a consequence, generally, higher-fatigue lives are found for the CLCF tests with the HT1 and HT2 compared to HT0. This general trend is not found for the TMF tests. However, only a low number of tests are used in this work so that the results are not yet significant for a more in-depth evaluation of the fatigue life behavior. Quite high strain amplitudes are considered in this work, which lead to low-fatigue lives of a few hundred cycles, but are necessary to obtain stress–strain hysteresis loops of the tool steel for modeling the cyclic viscoplastic behavior due to the high strength of the material. For HT0 and HT1 at lower temperatures, still rather narrow stress–strain hysteresis loops are measured which makes the determination of unique material properties more difficult. This is why the individual material properties for each HT and temperature are determined by using an experience-based approach in which material property values are defined from experimental evidence and some "manual" iterations. Numerical optimization methods can find unreasonable values if the material itself or the experimental results do not show the relevant phenomena described by the model to a sufficient extent [51]. Hence, they are not used here. Moreover, physically reasonable temperature dependencies are often not obtained when using optimization methods. Due to the applied experience-based approach, different persons might find different values, as documented in this work for the material properties. However, with the material properties determined in this work on the basis of the CLCF tests containing different strain amplitudes, different strain rates and dwell times, the cyclic plastic behavior and the increasing viscous behavior (strain rate dependency and stress relaxation) with increasing temperature can be described well. Moreover, these material properties allow a good description of TMF conditions as well.

Due to the relatively high complexity of the dependencies of the material properties on temperature, HT (thermal softening) and cyclic softening during fatigue loading, a stepwise procedure is proposed to determine the corresponding material properties and dependencies. Initially, when using individual material properties for HT0, HT1, and HT2, this results in a total of 120 material properties which are defined at the temperature sample points 20, 400, 500, 600, and 650 ${}^{\circ }$C. By introducing functional dependencies based on the individually determined material properties, the number of material properties is reduced to 24 with only a minimal impact on the results. It can be assumed that the same functional dependencies derived in this work also apply to other hot work steels. Hence, these dependencies could be directly used to fit the parameters of the used functions for a database consisting of CLCF tests that allow to gain the relevant information on the cyclic viscoplastic behavior. Two functions are used to describe the dependency of material properties on the carbide size *z* that is computed from a kinetic model derived for the martensitic tool steel X38CrMoV5-3 in [30]. The first function is a hyperbola and is reasonable, especially for strength-related material properties which, according to the Orowan mechanism, are expected to show the 1/z dependency. The second function describes a linear dependency on *z*. In contrast, the dependencies of the material properties on an aging variable used in the viscoplastic model of [35] to describe softening of a hot work steel are all linear. The use of linear dependencies for all material properties would have a negative influence on the model results in this work.

### 5.2. Thermal-Softening and Cyclic-Softening Model

The kinetic model used to describe the effect of thermal softening assumes that one carbide species exists that dominantly determines the mechanical properties [30]. Works on the microstructure of martensitic hot work steels have shown that various carbides precipitate in different sizes during the heat treatment [52,53]. Furthermore, the sizes of secondary carbides, namely MC and M${}_{2}$C, are observed at approximately one nanometer, respective to the size of M${}_{23}$C${}_{6}$ at approximately seven nanometers after austenitization and tempering [52]. For delimitation from a certain carbide species, a normalized carbide size is introduced in this work as a qualitative measure representing thermal softening. Furthermore, recording tempering-hardness curves is sufficient for calibration of the kinetic model, and thus no microscopic investigations are needed. A practical transfer to different steels containing secondary carbides is possible with the normalized carbide radius and makes the kinetic model more practical.

However, different secondary carbides which were, e.g., found in [52] could be considered individually in the kinetic model. Therefore, the sizes and their individual strengthening and coarsening behavior could be included in the kinetic model. However, the model complexity and the amount of material properties would be significantly higher than it is in the presented approach with one representative normalized carbide size. Moreover, for a full description a distinction between thermal- and plasticity-induced coarsening of the carbides would be necessary in the kinetic model. Plasticity-induced softening is, for example, explained by dislocations that can cut through the secondary carbides.

In comparison to the commonly used model of isotropic softening from Equation (Equation 5), the advantage of the cyclic-softening model developed in this work is visible in TMF tests. The new cyclic softening model considers history effects, so that cyclic softening occurring at higher temperatures also affects the behavior of a subsequent loading at lower temperature as it is observed in the TMF tests. This is confirmed by Figure 9 and Figure 10 when comparing the model results on the left (new model) and the right (standard model) side. However, for calibration of the cyclic softening model, TMF data must be considered to determine the maximal loss in strength that the material can suffer. Because TMF data was not used in other works on the softening of hot work steels, e.g., in [28,35,37], the relevance of the new softening model could not be identified there.

The cyclic softening model gives an exponential dependency of the softening variable *s* on accumulated plastic strain when integrating the evolution Equation (Equation 6) assuming isothermal conditions (see Equation (Equation 9)). This is appropriate for short-time softening. However, the experimental results show superimposed linear softening in the cyclic part of the isothermal tests, especially for the tests for HT2. Linear long-time softening is also observed in the OP TMF tests. Therefore, a model extension by a linear part as in Equation (Equation 5) could lead to an improvement of the cyclic softening model. To this end, the softening variable could be split additively into two variables: the first variable showing the exponential evolution from Equation (Equation 6) and the second variable a linear evolution or also an exponential evolution with small transition constant. The latter would allow to define saturation values so that the softening variable is a priori bound to the admissible range from 0 to 1.

In [37], it is shown that the softening behavior of a martensitic forging tool steel depends on the strain amplitude. All CLCF tests in this publication are executed with the same strain amplitude. Hence, dependency on the strain amplitude is not considered in this work. A corresponding extension of the model proposed in this work is, however, possible.

In [21], damage mechanisms of the hot work steel X38CrMoV5-3 are studied in thermal cycling experiments. It is found that in addition to thermal softening due to coarsening of the carbides, softening due to plastic straining has a major impact on the damage behavior. This is also observed in this work regarding the mechanical properties. Accordingly, thermal softening and cyclic softening are both included in the viscoplasticity model of this work. In [23], hot work steels were experimentally investigated. There, for isothermal fatigue tests, it is concluded that the short-time cyclic softening is driven by dislocations, e.g., annihilation and rearrangement of dislocations, whereas carbide coarsening is responsible for the long-time softening behavior. However, the fatigue tests are carried out at 450 and 550 ${}^{\circ }$C and, thus, the temperatures are lower than in this work. For temperatures up to 500 ${}^{\circ }$C in isothermal fatigue tests of this work, a strong impact of carbide size, i.e., thermal softening is observed. This is justified by the different maximum stress values observed for HT0, HT1, and HT2 in the initial cycles (see Figure 2). Hence, it can be assumed that the short-time softening is driven by thermal softening in contrast to the findings in [23]. This is also evident in the comparisons of the softening parts in Figure 13. However, only small plasticity is observed for example at 20 ${}^{\circ }$C (Figure 2) and, thus, dislocation rearrangement and annihilation might not occur in the tests at 20 and 400 ${}^{\circ }$C. The long-time softening is driven by cyclic softening which is linked to dislocation rearrangement and annihilation (see Figure 13). However, long-time softening was hardly observed for isothermal fatigue tests at 20 ${}^{\circ }$C (Figure 13) due to small plasticity in these tests. For elevated temperatures, e.g., 600 and 650 ${}^{\circ }$C, the short-time softening is driven by thermal and cyclic softening in common. In these tests, the amount of plasticity within a cycle is higher compared to the tests at 20 and 400 ${}^{\circ }$C so that dislocation rearrangement and annihilation might occur in the tests.

### 5.3. Microstructure-Related Aspects

The developed viscoplasticity model, including thermal and cyclic softening, is a phenomenological model whose material properties can be determined from macroscopic tests. Although softening has a strong relation to the microstructure of the martensitic tool steel, microscopic investigations are not considered in this work for validation of the model because the basic mechanisms underlying softening are assumed to be well understood from SEM and/or TEM investigations (e.g., [22,26,27,54]). Hence, with the developed model and the defined set of material tests for determination of the corresponding material properties, a practical model is now available suited for application.

In addition to coarsening of secondary carbides, an evolution in the martensitic material matrix can be observed for hot work steels in high-temperature conditions. Prior austenite grains solidify to martensitic laths as the main structure of the material. This suggests that the evolution of martensitic laths also has an impact on the material’s strength. It is shown in [55] that carbon-rich interlath phases containing of retained austenite are within the martensitic laths. Because there are high contents of carbon in these interlath phases, this carbon is missing for the precipitation of secondary carbides. The size of interlath phases depends on the cooling rate. Thus, it is difficult to develop a model with wide application range which considers martensitic laths and the precipitates, i.e., secondary carbides, in detail. The change of martensitic laths in cyclic thermal tests was investigated in [56]. Overall, the microstructural changes could be implemented in the kinetic model; however, the focus of the presented viscoplasticity model in this work is on TMF behavior and, thus, a macroscopic viscoplasticity model is chosen.

### 5.4. Future Works

Additive manufacturing has become more important for hot work tools. Recently, the material X38CrMoV5-3 for which the viscoplasticity model was developed is manufactured also by additive manufacturing such as laser melting. Current works investigate the material behavior which differs from conventional manufactured tools, e.g., [57,58,59,60,61]. In future investigations, thus, an additive-manufactured specimen could be tested and compared with results computed with the viscoplasticity model. Thereby, deviations could be identified and coupled with the manufacturing process.

## 6. Conclusions

In this paper, a viscoplasticity model is developed for the martensitic hot work steel X38CrMoV5-3 based on CLCF and TMF tests in the temperature range from 20 to 650 ${}^{\circ }$C. The model considers thermal as well as cyclic softening. A new softening variable and a corresponding evolution equation are introduced. The results of the tests performed for material with different heat treatments are used to determine the material properties of the model. To this end, a stepwise procedure is proposed. The results are concluded as follows.

The experimental results of the CLCF and TMF tests show a significant effect of heat treatment, i.e., thermal softening, as well as cyclic softening during the test on the mechanical properties, especially in the CLCF tests at higher temperatures and the TMF tests. The designed test program has proven to be efficient for the determination of the material properties of the proposed viscoplasticity model.The investigated steel has wide application in processes where heat resistance, hardness, and heat toughness are required. The model can be transferred to materials that show the same phenomena under TMF, e.g., steels in power-generation applications. An efficient determination of the material properties on basis of experimental results is possible due to the phenomenological modelling approach.A new cyclic-softening model is derived that describes history effects found during thermomechanical loading. The cyclic-softening model describes the evolution of a softening variable *s* for isothermal and thermomechancial conditions.A stepwise, experience-based approach is presented to determine the material properties and their functional dependency on the size of secondary carbides controlling thermal softening based on the isothermal CLCF tests. For the determination of the material properties of the new cyclic-softening model, the results of the TMF tests showing the history effect needs to be employed. A calibration of the model without TMF tests is, hence, not possible.The viscoplasticity models and the determined temperature-dependent material properties give a good overall description of the complete data from CLCF and TMF tests with different heat treatments.A three-dimensional formulation of the viscoplasticity model can be obtained by using the von Mises yield criterion with kinematic hardening and is well suited for finite-element implementation to assess the thermomechanical behavior and fatigue life of hot work tools.

## Figures and Tables

**Figure 1 materials-16-00994-f001:**
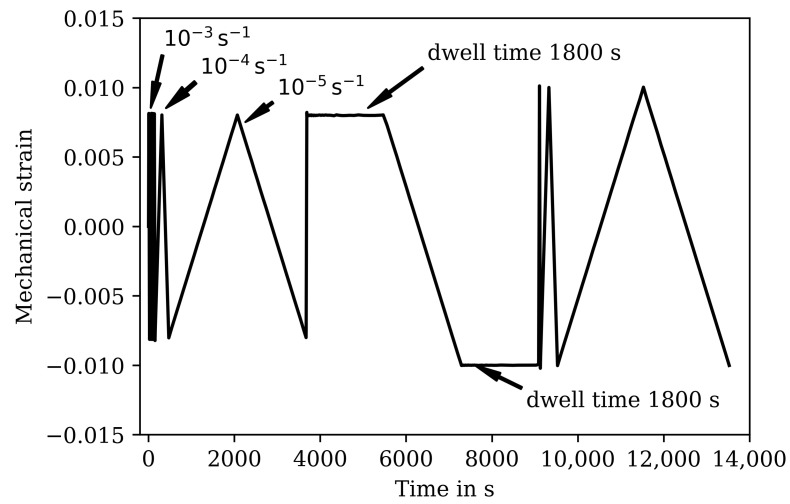
Mechanical strain over time for the complex cycles of the CLCF tests with different strain rates and dwell times.

**Figure 2 materials-16-00994-f002:**
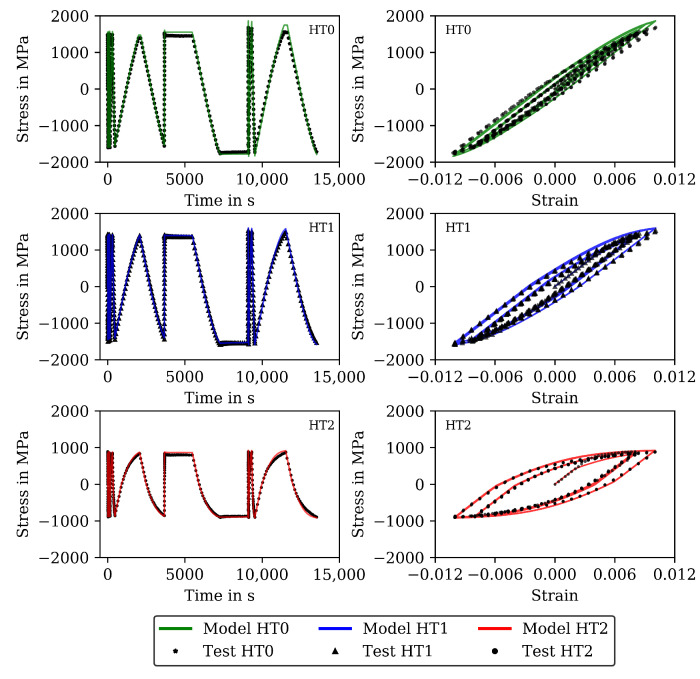
Measured stress and stress calculated with the model of step 1 for the initial (11) cycles of the CLCF tests at 20 ${}^{\circ }$C for the HT0, HT1, and HT2 material. Stress-time diagrams (**left**) and stress–strain hysteresis loops (**right**).

**Figure 3 materials-16-00994-f003:**
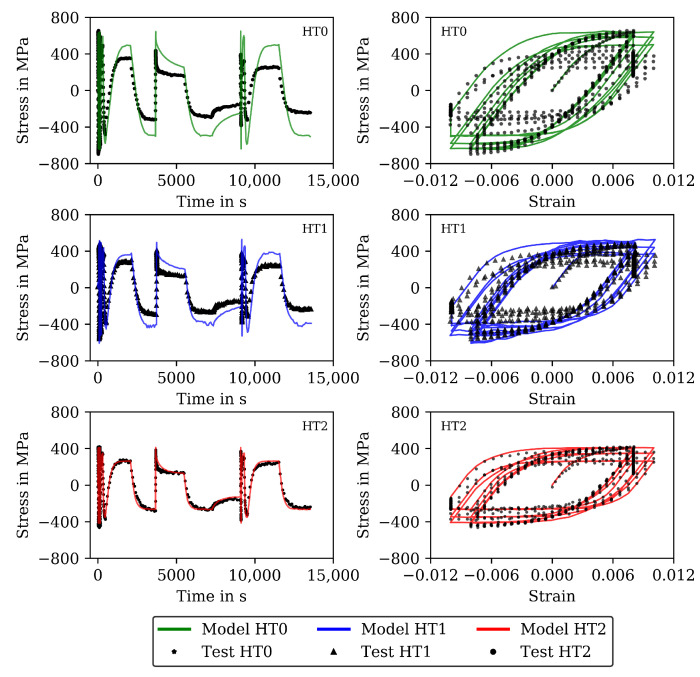
Measured stress and stress calculated with the model of step 1 for the initial (11) cycles of the CLCF tests at 650 ${}^{\circ }$C for the HT0, HT1 and HT2 material. Stress-time diagrams (**left**) and stress–strain hysteresis loops (**right**).

**Figure 4 materials-16-00994-f004:**
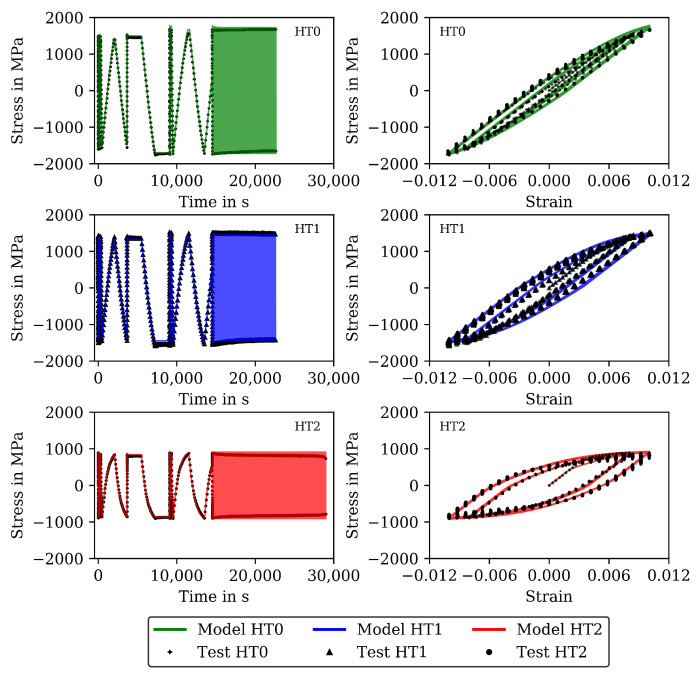
Measured stress and stress calculated with the model of step 2 for all cycles to failure of the CLCF tests at 20 ${}^{\circ }$C for the HT0, HT1, and HT2 material. Stress–time diagrams (**left**) and stress–strain hysteresis loops (**right**).

**Figure 5 materials-16-00994-f005:**
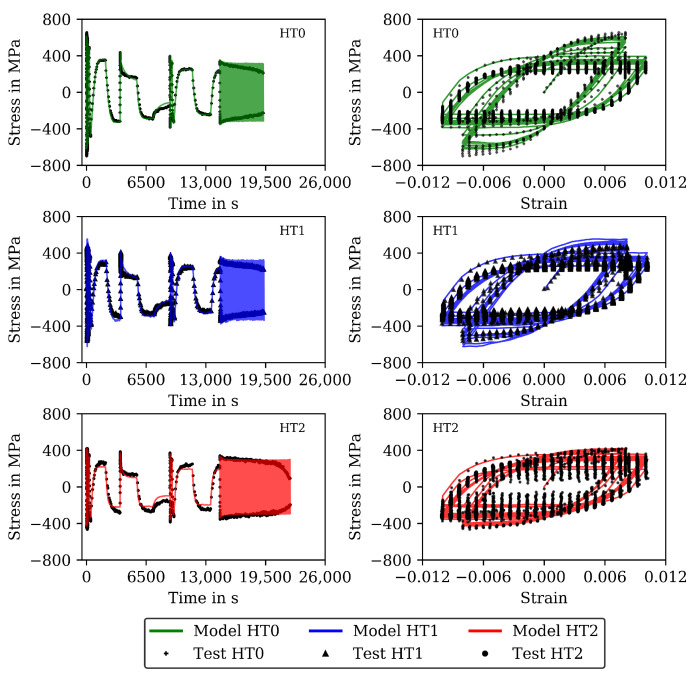
Measured stress and stress calculated with the model of step 2 for all cycles to failure of the CLCF tests at 650 ${}^{\circ }$C for the HT0, HT1 and HT2 material. Stress–time diagrams (**left**) and stress–strain hysteresis loops (**right**).

**Figure 6 materials-16-00994-f006:**
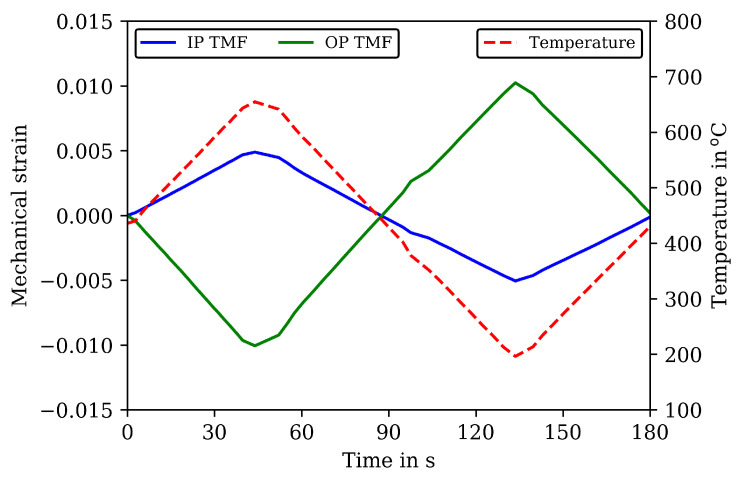
Mechanical strain and temperature versus time in in-phase (IP) and out-of-phase (OP) TMF tests.

**Figure 7 materials-16-00994-f007:**
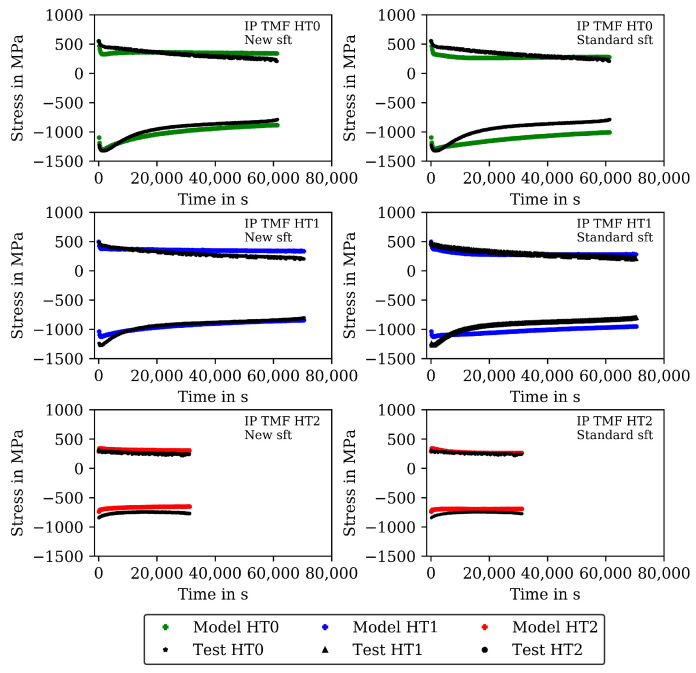
Stress–time diagrams for the measured stress and stress calculated with the viscoplasticity model for the in-phase TMF tests for the HT0, HT1, and HT2 material at the points of strain reversal (mechanical strain amplitude of 0.005). New isotropic-softening model (**left**) and standard isotropic-softening model (**right**).

**Figure 8 materials-16-00994-f008:**
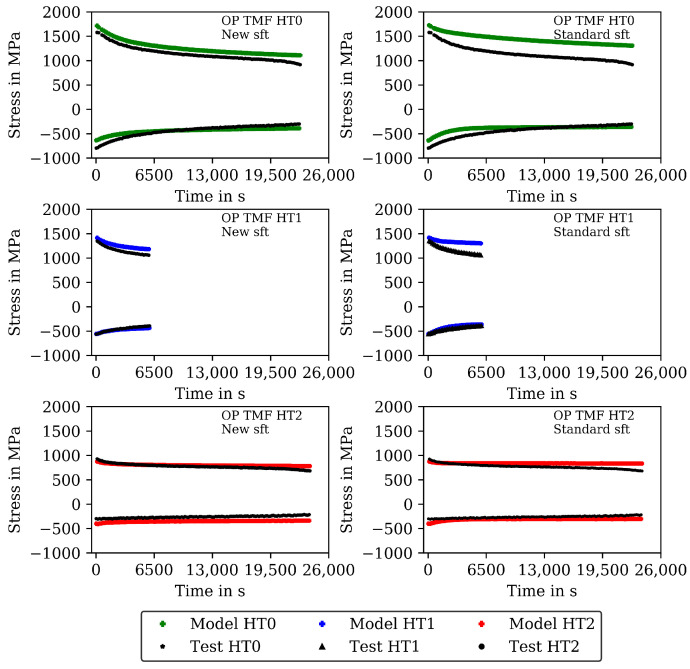
Stress–time diagrams for the measured stress and stress calculated with the viscoplasticity model for the out-of-phase TMF tests for the HT0, HT1, and HT2 material at the points of strain reversal (mechanical strain amplitude of 0.01). New isotropic-softening model (**left**) and standard isotropic-softening model (**right**). Incomplete data for HT1 due to interrupted data recording.

**Figure 9 materials-16-00994-f009:**
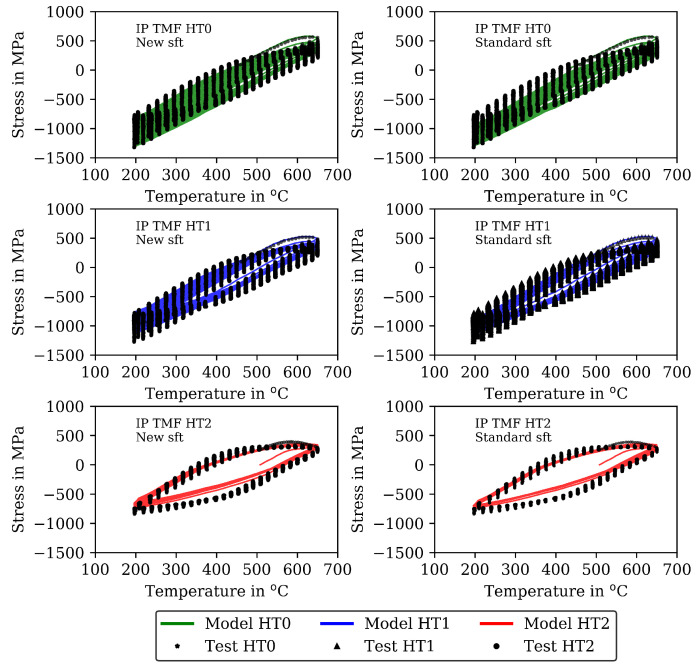
Stress–temperature diagrams for the measured stress and stress calculated with the viscoplasticity model for the in-phase TMF tests for the HT0, HT1, and HT2 material (mechanical strain amplitude of 0.005). New isotropic-softening model (**left**) and standard isotropic-softening model (**right**).

**Figure 10 materials-16-00994-f010:**
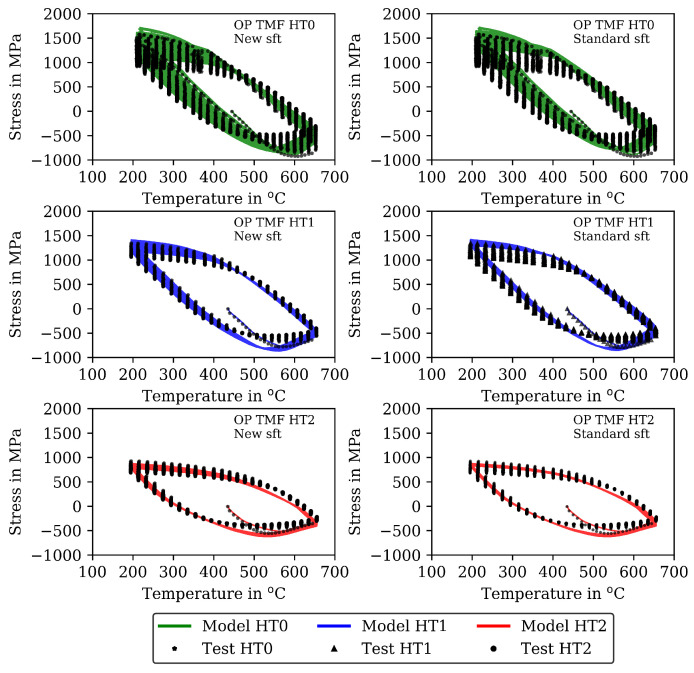
Stress–temperature diagrams for the measured stress and stress calculated with the viscoplasticity model for the out-of-phase TMF tests for the HT0, HT1, and HT2 material (mechanical strain amplitude of 0.01). New isotropic softening (**left**) and standard isotropic softening (**right**). Incomplete data for HT1 due to interrupted data recording.

**Figure 11 materials-16-00994-f011:**
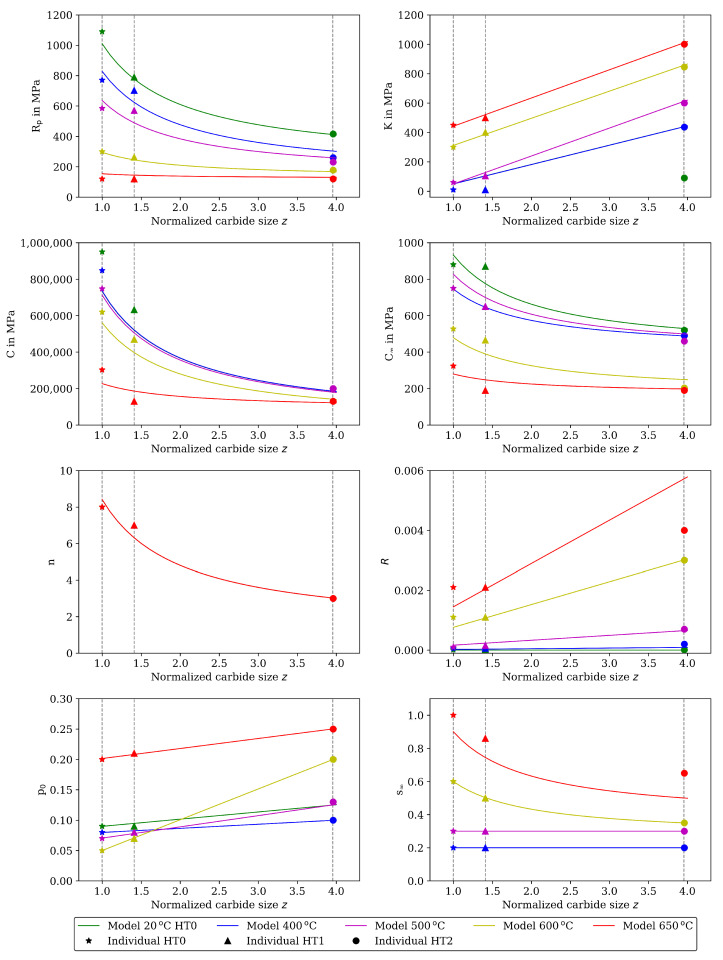
Material properties of the viscoplasticity model versus carbide size for different heat treatments, i.e., carbide sizes and temperatures. Individual determined material properties of step 1 are presented as symbol, functional dependencies of step 2 as line.

**Figure 12 materials-16-00994-f012:**
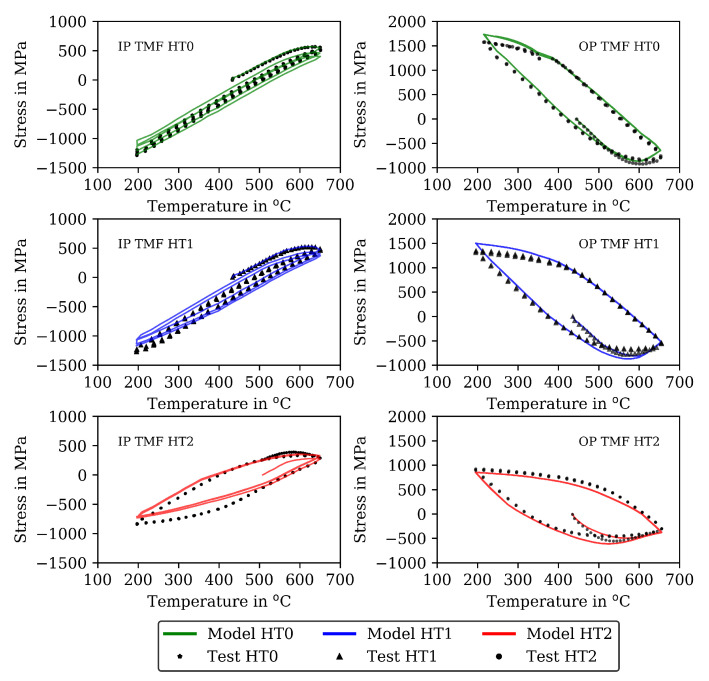
Measured stress and stress calculated with the model of step 1 for the first three cycles of the TMF tests for the HT0, HT1, and HT2 material: In-phase TMF stress–temperature hysteresis loops (mechanical strain amplitude of 0.005, **left**) and out-of-phase TMF stress–temperature hysteresis loops (mechanical strain amplitude of 0.01, **right**).

**Figure 13 materials-16-00994-f013:**
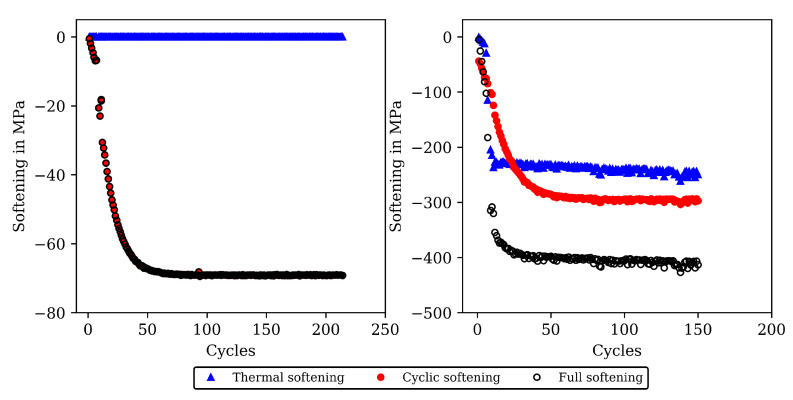
Softening that is lost by the different softening proportions (thermal softening, cyclic softening, and full softening) of the viscoplasticity model from step 2. The softening proportions are subtracted from the model results of stress at the load reversal points without softening. The results are presented for 20 ${}^{\circ }$C (**left**) and for 650 ${}^{\circ }$C (**right**), both for the CLCF-tests for the HT0 material.

**Figure 14 materials-16-00994-f014:**
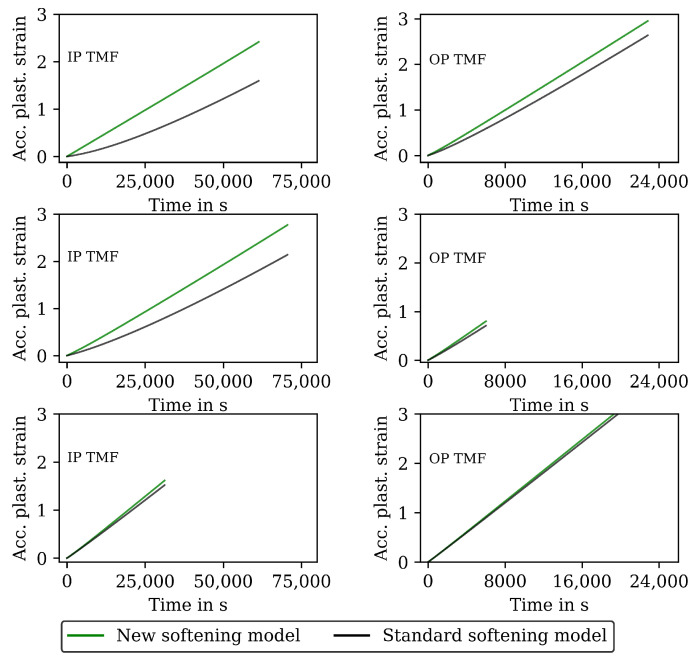
Accumulated plastic strain calculated with the new cyclic-softening model (**left**) and the standard isotropic-softening model (**right**) for the in-phase and out-of-phase TMF tests for the HT0 material.

**Table 1 materials-16-00994-t001:** Chemical composition of the specimen and chemical composition in accordance to DIN EN ISO 4957 in wt%, Fe-bal.

	C	Si	Mn	P	S	Cr	Mo	V
Measured	0.388	0.49	0.40	0.021	0.004	5.15	2.60	0.504
Min.	0.35	0.3	0.3	-	-	4.8	2.7	0.4
Max.	0.4	0.5	0.5	0.03	0.02	5.2	3.2	0.6

**Table 2 materials-16-00994-t002:** Heat treatment time and temperature of the specimen.

Notation	Heat Treatment
HT0	initial state
HT1	300 min. at 600 ${}^{\circ }$C
HT2	1000 min. at 650 ${}^{\circ }$C

**Table 3 materials-16-00994-t003:** Overview of the CLCF tests with the number of cycles to failure.

Temperature	20 ${}^{\circ }$C	400 ${}^{\circ }$C	500 ${}^{\circ }$C	600 ${}^{\circ }$C	650 ${}^{\circ }$C
Number of cycles to failure HT0	215	153	133	69	130
Number of cycles to failure HT1	213	197	275	141	150
Number of cycles to failure HT2	373	237	250	251	203

**Table 4 materials-16-00994-t004:** Overview of the in-phase and out-of-phase TMF tests with the number of cycles to failure.

Type	In-Phase TMF	Out-of-Phase TMF
Number of cycles to failure HT0	341	127
Number of cycles to failure HT1	392	155
Number of cycles to failure HT2	174	133

## Data Availability

The material properties presented in this study are available in the Appendix A. The subroutines and experimental data used in this study are available on request from the corresponding author.

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
