# Peer review of "A Temperature-Dependent Viscoplasticity Model for the Hot Work Steel X38CrMoV5-3, Including Thermal and Cyclic Softening under Thermomechanical Fatigue Loading"

_materials, 2023, doi:10.3390/ma16030994_

Round 1

Reviewer 1 Report

This work presented a viscoplasticity model describing the softening behavior under thermomechanical fatigue loading. One problem should be clarified to further improve the quality of this paper. The deviation between the model and test results should be explored by introducing the detailed microstructural changes of samples.

Reviewer 2 Report

The manuscript is devoted to the development of a model of viscoplasticity depending on temperature, which describes the thermal and cyclic softening of martensitic hot work steel X38CrMoV5-3 under thermomechanical fatigue loading. The authors made a large number of experiments. The article is well written, the results are well presented and thoroughly discussed. As a result, the authors made significant conclusions and developed a new cyclic softening model that describes the history effects found during thermomechanical loading.

I have a few small comments:

1. Material. “…Tab. 1 shows the chemical composition in accordance to DIN EN ISO 4957…”. Why do the authors not indicate the exact chemical composition of the alloy?

2. Material. “…Tab. 2 shows the different tempering procedures with the assigned notations HT0, HT1 and HT2….” Why were these heat treatment regimes (temperatures and times) chosen? I think the authors should explain this.

3. “All tests are performed on an electromechanical fatigue testing system of Instron in strain control by measuring the strain with an Epsilon extensometer.” Please identify the test system model.

Reviewer 3 Report

The article is scientific, thematically suitable for the journal.

I have the following comments regarding the presentation of the manuscript:

- It is necessary to add the Fe content to Table 1 as it is steel.

- In the description of tab. 1 is the chemical composition in % (in what % atm. or wt.?)

- Fig. 1 in what units is Mechanical strain?

- All test and measuring devices used in the experiment should be defined in the material and methods. Their (manufacturer, state, accuracy, etc.).

- As part of the implementation of experiments in the field of physical metallurgy, precipitation processes, spheroidization - carbide formation, it would be appropriate, especially for medium and high-alloyed steels, to specify a specific type of material with a precisely defined chemical composition.

- In your case, you indicate the intervals of chemical contents of the given type of steel defined by the material sheet (standard) from min. after max. It is therefore advisable to work with exact values during calculations.

- In the introduction, the novelty, the main contribution of the experiment, should be clearly defined. What fundamental will it bring and it is still unknown. Similarly, in the conclusions, it is appropriate to define what has been discovered.

The research is extensive, well processed in terms of the presentation of measured values. How you write plays an important role, but the structure of the evaluated materials also plays a role. Microstructural analysis of the influence of individual processes should also be evaluated and presented in such a focused experiment. You present the changes taking place in martensitic structures only on the basis of references to other literature. It is fine, but due to the complexity of the research, it would be appropriate to carry out these structural analyzes on the samples evaluated by you.

Reviewer 4 Report

Strengths

In this paper, a viscoplasticity model is developed for the martensitic hot work steel X38CrMoV5-3 based on CLCF and TMF tests in the temperature range from 20 to 650 C. The model considers thermal as well as cyclic softening. A new softening variable and a corresponding evolution equation are introduced. The results of the tests performed for material with different heat treatments are used to determine the material properties of the model. To this end, a step-wise procedure is proposed. The results are concluded as follows:

• The experimental results of the CLCF and TMF tests show a significant effect of heat treatment, i.e. thermal softening, as well as cyclic softening during the test on the mechanical properties, especially in the CLCF tests at higher temperatures and the TMF tests.

• The investigated steel has wide application in processes where heat resistance, hardness and hot toughness are required. The model can be transferred to materials that show the same phenomena under TMF, e.g. steels in power generation applications. An efficient determination of the material properties on basis of experimental results is possible due to the phenomenological modelling approach.

• A new cyclic softening model is derived that describes history effects found during thermomechanical loading. The cyclic softening model describes the evolution of a softening variable s for isothermal and thermomechancial conditions.

• A step-wise experience-based approach is presented to determine the material properties and their functional dependency on the size of secondary carbides controlling thermal softening based on the isothermal CLCF tests. For the determination of thematerial properties of the new cyclic softening model, the results of the TMF tests showing the history effect needs to be employed. A calibration of the model without TMF tests is, hence, not possible.

• The viscoplasticity models and the determined temperature dependent material properties give a good overall description of the complete data from CLCF and TMF testswith different heat treatments.

• A three-dimensional formulation of the viscoplasticity model can be obtained using the von Mises yield criterion with kinematic hardening and is well suited for finite element implementation to assess the thermomechanical behavior and fatigue life of hot work tools.

Weakness    

1. Citation are not filled in (Line 19).

2. Table 1, not Tab. 1 (Line 152, 158, 179 and others).

3. Figure 1, not Fig. 1 (Line 171, 181,182 and others).

4. Equation (3), not eq. (3) (Line 253).

5. Reference to Figure 7 after his appearance in the text (Line 466).    6. Five publications [25, 30, 36, 45, 50] of only one of the authors in the links.

Round 2

Reviewer 1 Report

The authors have carefully revised the manuscript. This manuscript can be accepted.